# Effect of the Fat Eaten at Breakfast on Lipid Metabolism: A Crossover Trial in Women with Cardiovascular Risk

**DOI:** 10.3390/nu12061695

**Published:** 2020-06-06

**Authors:** Jessica M. Delgado-Alarcón, Juan José Hernández Morante, Francisco V. Aviles, María D. Albaladejo-Otón, Juana M. Morillas-Ruíz

**Affiliations:** 1Health Science PhD program, Catholic University of Murcia, 30107 Murcia, Spain; jmdelgado@alu.ucam.edu; 2Eating Disorder Research Unit., Catholic University of Murcia, 30107 Murcia, Spain; 3Service of Biochemistry, Hospital Universitario Virgen de la Arrixaca, 30120 Murcia, Spain; faviles@yahoo.es; 4Service of Biochemistry, Hospital Universitario Santa Lucia of Cartagena, 30202 Murcia, Spain; mariadoalbaladejo@gmail.com; 5Food Technology and Nutrition Department, Catholic University of Murcia, 30107 Murcia, Spain

**Keywords:** fatty acid, breakfast, lipoprotein, apolipoprotein, cardiovascular risk

## Abstract

Recent studies point out that not only the daily intake of energy and nutrients but the time of day when they are ingested notably regulates lipid metabolism and cardiovascular risk (CVR). Therefore, the aim of the study was to assess if the type of fat ingested at breakfast can modify lipid metabolism in women with CVR. A randomized, crossover clinical trial was performed. Sixty volunteers were randomly assigned to a (A) polyunsaturated fatty acid (PUFA)-rich breakfast, (B) saturated fatty acid (SFA)-rich breakfast, or (C) monounsaturated fatty acid (MUFA)-rich breakfast. Plasma lipoprotein and apolipoprotein subfractions were determined. Our data showed that the PUFA-rich breakfast decreased lipoprotein (a) (Lp(a)), very low-density lipoproteins (VLDL), and intermediate-density lipoproteins (IDL), and increased high-density lipoproteins (HDL). A similar trend was observed for the MUFA-rich breakfast, whereas the SFA-rich breakfast, although it decreased VLDL, also increased IDL and reduced HDL. The PUFA-rich breakfast also decreased β-lipoproteins and apolipoprotein-B. In summary, varying the type of fat eaten at breakfast is enough to significantly modify the lipid metabolism of women with CVR, which can be of great relevance to establish new therapeutic strategies for the treatment of these subjects.

## 1. Introduction

Apart from specific situations like the current COVID-19 epidemic, cardiovascular diseases (CVD) continue to be the leading cause of mortality worldwide [1]. Many advances have been made in this regard, and it is now possible to determine the risk of cardiovascular disease. Thus, risk prediction scores for cardiovascular disease (CVD) contain information on total cholesterol and high-density lipoprotein cholesterol (HDL), among other conventional risk factors [2]. Considerable interest exists on whether CVD prediction can be improved by assessment of additional lipid-related markers, such as several proatherogenic lipoprotein subfractions (small dense low-density lipoprotein (LDL) cholesterol or apolipoprotein B (ApoB) [3,4,5]. In contrast, other markers, especially HDL and apoprotein ApoA-I, have been shown to have anti-inflammatory effects in various clinical studies in which biomarkers can better predict organ failure [6].

Several recent studies have shown independent relationships between levels of LDL and HDL subclasses and risk of both coronary artery and cerebrovascular disease [7,8]. Also, there is inconclusive evidence suggesting that LDL and HDL subfractions measurement improves clinical assessment of CVR [9]. Studies suggest that more refined analyses of lipoprotein subspecies may lead to further improvements in CVR evaluation, since the characterization of LDL subclasses, in particular regarding lipoprotein(a) (Lp(a)) and small dense LDL, may help to develop more personalized interventions focused on individual characteristics [10,11].

It is well established that diet, or more specifically the fatty acid composition of the diet, is one of the main factors involved in the regulation of plasma lipid profile [12,13]. The relationship between diet and blood cholesterol or lipid markers is complex; many different dietary components can affect cholesterol levels (proteins, carbohydrates, plant sterols, etc.). But the most controversial component is dietary fat, both regarding the quantity and the fatty acid composition [14,15].

Recent findings show that not only is the total amount of fat important, but also the time of day when they are ingested [16]. In this line, several previous reports have shown the importance of adjusting the diet to the subject’s biological rhythms. In fact, two common mistakes have been described that can upset the balance of the diet despite adequate nutrient intake. On the one hand, eating too late, and more specifically after 15 PM, reduces the efficacy of a hypocaloric dietary treatment and increases the risk of obesity development [17]. On the other hand, skipping breakfast can also predispose to developing obesity and other associated disorders [18,19,20], which highlights the importance of this intake, even being considered as the most important intake of the day [21,22]. However, to the best of our knowledge, no work has been conducted that assesses whether the type of fat ingested at breakfast has any effect on lipoprotein parameters related to CVR.

Therefore, the purpose of the present study was to evaluate whether changing exclusively the fat ingested at breakfast is enough to modify the lipoprotein and apolipoprotein subclasses for women with CVR. In addition, through the present crossover trial, a secondary objective was to determine the best fatty acid composition at breakfast to deliver a better lipoprotein and apolipoprotein profile regarding cardiovascular risk parameters.

## 2. Materials and Methods

### 2.1. Study Design

A longitudinal, crossover trial was designed to evaluate the effect of three types of fats eaten at breakfast on plasma lipoproteins, lipoprotein subfractions, and apoproteins. Breakfast compositions were: breakfast-1 (polyunsaturated fatty acids (PUFA) source, containing 20 g of margarine with no trans fat), breakfast-2 (saturated fatty acids (SFA) source, containing 20 g of butter), and breakfast-3 (monounsaturated fatty acids (MUFA) source, containing 20 g of virgin olive oil). The lipid composition of the three types of fats is available in Figure 1 (further information in Appendix A). All participants performed each of the three identical experimental procedures for 30 days, with a 45-day washout period between interventions. The subjects were randomized in alternating sequence by a Latin square procedure, which was previously designed. A researcher (J.J.H.M.) carried out the randomization with the assistance of a macro designed in Visual Basic for the Microsoft Excel program.

The study was carried out after receiving written authorization from the Ethics Committee of the Catholic University of Murcia (N#1874). Participants were informed, either orally or in writing, about the study design. They were also given an explanation of the ethical aspects of the project, informing the patients about the main objective of the study and guaranteeing the confidentiality and anonymity of the data, in accordance with the Declaration of Helsinki and Biomedical Research Spanish Law. All participants provided written informed consent. This trial was registered at Australian New Zealand Clinical Trials Registry (ACTRN) (further information available at: http://www.ANZCTR.org.au/ACTRN12611000456954.aspx. Code: #12611000456954).

### 2.2. Participants

A total of 60 women volunteers gave informed consent to take part in the present study. Mean age and body mass index (BMI) were 63.5 ± 18.4 y and 27.79 ± 3.97 kg/m^2^, respectively. To increase the treatment adherence and reduce bias, the target population consisted of subjects who lived in the same institution, with similar lifestyle habits and with identical daily meals, as previously recommended [23]. Those subjects with significant cognitive impairment, psychiatric disorders, or chronic pharmacological treatment that may have affected the effectiveness of the dietary intervention (corticosteroids, thyroid hormones, oral antidiabetic agents, or lipid-lowering drugs) were excluded. A chronic disease that may have interfered with dietary therapy (cancer, renal or hepatic impairment, chronic gastrointestinal conditions), or acute disease episodes during the study were also established as exclusion criteria. Those subjects who had followed a hypocaloric diet at the allocation time or in the three months prior to the beginning of the study were also excluded.

The necessary sample size was estimated with the assistance of the program GPower 3.0 (Dufsseldorf, Germany) [24]. The sample size was calculated according to our previous report [25], taking into account a confidence level (1−α) of 95% and considering a power (β) of 80%; we selected a difference of effect between groups or effect size (d) equal to a 5% reduction in the lipoprotein composition. The standard deviation (σ) selected was 10%, taking into account our previous study [25], resulting in 41 subjects per group. Considering an estimated drop-out rate of 5%, the final minimum sample for the present study was 45 subjects per group. Figure 2 shows the flow diagram followed for the patients’ recruitment and selection procedures.

### 2.3. Intervention

In addition to a serving portion of the fat source (margarine, butter, or virgin olive oil), each breakfast consisted of low-fat milk (200 mL), instant coffee (1 monodose sachet of 18 g), sugar (1 monodose sachet of 8 g), and two white bread toasts. During the washout period, breakfast fat was substituted by a pineapple juice (200 mL) and peach ham (50 g). Except the type of fat contributing to the breakfast, the nutritional composition of the other daily meals was identical (1636 ± 527 Kcal/day, 61 ± 23 g proteins/day, 203 ± 59 g carbohydrates/day, 65 ± 35 g fats/day). The diets were designed depending on the volunteers’ requirements and based on the volunteers’ nutritional habits to enhance adherence. At the beginning of the study, the volunteers were instructed to follow the assigned diet, without modifying their lifestyle (physical activity, sleeping habits, meal schedules, etc.) during the experimental period.

The intervention included five phases: three phases of dietary intervention, in which each volunteer was randomly assigned to ingest in a certain order each breakfast type for a 30-day period, and two washout periods (of 45 days each). The menu consumed by volunteers was designed for 30 consecutive days and was identical during the intervention, the type of fat consumed during breakfast being the only nutritional variable that was modified in the study.

Although fat content was slightly different within the different breakfasts, the aim of this study was to employ different fats contained in commercialized usual portions of each type of food (margarine, butter, and olive oil) to analyze if a normal fat ration at breakfast could modify the plasma lipoprotein profile, lipoprotein subfractions, and apoproteins. These single-dose formats are commercially available and frequently used, offered daily in hospitals, restaurants, hotels, business coffee, study centers, and so forth, and are consumed by a high percentage of the population at breakfast. Weight, arterial blood pressure, and cardiac frequency were measured on the first and last day (pre-/post-treatment) of each of the three breakfast periods in fasting conditions.

### 2.4. Lipid Metabolism Parameters

Blood samples were obtained, after 12 h fast, from the antecubital vein into 9 mL siliconized tubes. After venipuncture, samples were kept on ice and then centrifuged at 3500 rpm (Heraus Biofugue Stratos, Thermo Scientific, Dreieich, Germany) for 10 min at 4 °C. The mean time between venipuncture and centrifugation was 50 min (interquartile range: 30–70 min). Plasma samples were stored at −80 °C until further analyses.

Lipoprotein (chylomicrons, beta, pre-beta, alpha lipoproteins) were measured by lipidogram in agarose gel with Hydragel 7 Liprotein^®^ kit (Sebia, Lisses, France), following the manufacturer’s instructions. Briefly, the analysis was carried out by electrophoresis on buffered agarose gels (pH 7.5) on the HYDRASYS semiautomatic instrument. The procedure was based on the migration capacity of lipoproteins: chylomicrons normally remain at the point of application, beta lipoproteins migrate at the position of beta-2 globulins, the pre-beta lipoproteins migrate between beta and alpha-globulin, and finally, alpha lipoproteins are the fastest and migrate at the position of alpha-2 globulins. To carry out this procedure, 10 µL of plasma were placed in an agarose gel, and electrophoresis was performed. The resulting gel was placed in a paper film and stained with 220 mL of dye solution (Sudan Black). The Hydroscan 2 system (Sebia, Lisses, France) was employed to quantify the different lipoprotein subfractions. On each gel, a normal control was used to check the electrophoresis process and the staining procedures [26].

Apolipoproteins (A-I, A-II, E, B) were measured by immune-nephelometry with the BN ProSpec^®^ System (Siemens, Barcelona, Spain), according to the manufacturer’s guidelines. This automatic system allows the automatic determination of plasma proteins through the nephelometric system [27]. The result is evaluated by comparison with a standard of known concentration.

Lipoproteins subfractions (lipoprotein VLDL, IDL-C, IDL-B, IDL-A, LDL-1, LDL-2, LDL-3, LDL-4, LDL-5, LDL-6, LDL-7, and HDL) were measured by polyacrylamide gel electrophoresis using the Lipoprint kit from Quantimetrix^®^ (Redondo Beach, CA, USA). Each kit contains a preloaded polyacrylamide gel and a load buffer including a specific dye for lipoproteins. Briefly, 25 µL of plasma samples were loaded with 200 µL of buffer. Electrophoresis was then performed for 1 h, and the resulting gel was scanned with the equipment. The scanned images were analyzed with the NIH Image software (https://imagej.nih.gov/nih-image/about.html). Quantification was performed through the Quantimetrix^®^ software, which automatically identifies lipoprotein bands of the gel, based on their mobility (Rf). The following values were assigned: VLDL Rf:0.00-0-06, IDL-C Rf:0.13, IDL-B Rf:0.22, IDL-A Rf:0.27, LDL-1 Rf:0.33, LDL-2 Rf:0.30, LDL-3 Rf:0.44, LDL-4 Rf:0.49, LDL-5 Rf:0.54, LDL-6 Rf:0.58, LDL-7 Rf:0.74, and HDL Rf:0.74–1.00 [28]. The lipoprotein particles migrated through the separating gel matrix and were resolved into lipoprotein bands according to their particle sizes from: HDL (migrates the farthest), small dense LDL, larger buoyant LDL, midbands (comprising primarily IDL), and VLDL. Chylomicrons, if present, appear above the stacking gel or remain in the loading gel. A typical lipoprint profile consists of 1 VLDL band, 3 IDL midbands, and up to 7 LDL bands. After electrophoresis was completed, stained lipoprotein fractions (bands) were identified by their mobility (Rf) using VLDL as the starting reference point (VLDL = 0) and HDL as the leading point (HDL = 1).

### 2.5. Statistical Analysis

Statistical analysis was performed with SPSS 25 software (SPSS Inc., Chicago, IL, USA). Data are presented as mean ± standard deviation unless otherwise stated. Data distribution was determined using the Kolmogorov–Smirnov test. Differences between variables were tested with repeated measures analysis of covariance (ANCOVA), considering age, weight, and intervention order as covariates. To compare the effectiveness among the different interventions, the same statistical procedure was carried out considering the change (final-baseline) of the different variables as an estimation of the treatment effect. Bonferroni’s post hoc test was performed to avoid bias due to multiple comparisons. Statistical significance was set at *p* < 0.05.

## 3. Results

### 3.1. Clinical Characteristics of the Subjects

Figure 2 shows the details of participants’ allocation. Fifty-three women completed the study.

Their baseline characteristics are shown in Table 1. Mean baseline body mass index (BMI) was above the cut-off point considered as normal weight (24.9 kg/m^2^) and was therefore used as a covariate in the analyses of the intervention effects. Seven women were previously diagnosed with type 2 diabetes, while nine women had personal history of cardiovascular disease. Regarding baseline lipoprotein values, both low-density lipoprotein (LDL) and high-density lipoprotein (HDL) were within the normal range.

### 3.2. Effect of the Breakfast Fat on Lipoprotein Subfraction

The effects of the different interventions on the lipoprotein subfraction parameters are shown in Figure 3 and Appendix A. Main effects were observed regarding HDL (Figure 3d), which was significantly increased both with the polyunsaturated fatty acid (PUFA)-rich (*p* < 0.001) and the monounsaturated fatty acid (MUFA)-rich (*p* = 0.017) breakfasts; in contrast, the saturated fatty acid (SFA)-rich breakfast induced a 7% reduction of this lipoprotein (*p* = 0.002). The PUFA-rich breakfast also showed a statistically significant reduction of lipoprotein(a) (Lp(a)), very low-density lipoprotein (VLDL), and total intermediate-density lipoprotein (IDL), mainly due to the reduction of the IDL-c subfraction, as well as LDL5 (Appendix A). Unlike in the previous case, the SFA-rich breakfast induced an increase of IDL-b and IDL-c, so total IDL was also increased, but interestingly, this treatment also induced a significant reduction of VLDL (Figure 3a). Meanwhile, the MUFA-rich breakfast reduced most of the lipoproteins, but the statistical significance was only reached for HDL, as commented. Specific mean values and statistical significance are described in Appendix A.

When lipoprotein subfractions were evaluated attending to their electrophoretic mobility (Figure 4), our data showed that PUFA-rich and MUFA-rich breakfasts were able to reduce chylomicrons (*p* = 0.021 and *p* = 0.003, respectively, Figure 4a). Whereas PUFA-rich breakfasts caused a statistically significant reduction of β-lipoprotein (*p* < 0.001), both SFA-rich and MUFA-rich breakfasts were associated with an increase of this subfraction (*p* = 0.025 and *p* < 0.001, respectively) (Figure 4b). The opposite situation was observed regarding the preβ-lipoprotein, since PUFA-rich breakfasts increased this fraction (*p* = 0.017) but SFA-rich breakfasts reduced the plasma values of this fraction (*p* < 0.001) (Figure 4c). The lipoproteins that migrate to the α-position were increased with the PUFA-rich breakfast (*p* = 0.018); in contrast, the MUFA-rich intervention was characterized by a reduction of this fraction (*p* = 0.007) (Figure 4d).

### 3.3. Effect of the Breakfast Fat on Apolipoproteins

The data derived from apoliproteins (Apo) are shown in Figure 5. The PUFA-rich breakfast produced an increase of apolipoprotein ApoAI (*p* < 0.001) and ApoE (*p* < 0.001) (Figure 5a,d), whereas the SFA-rich breakfast had the opposite effect, as there was a decrease in ApoAI (*p* = 0.004) and ApoAII (*p* < 0.001) (Figure 5a,b). Moreover, the MUFA-rich breakfast produced a drop in ApoAII levels (*p* = 0.001) (Figure 5b). Specific mean values and statistical significance are described in Appendix A.

## 4. Discussion

The objective of the present work was to evaluate if the modification of fat intake only at breakfast was enough to modify lipid/lipoprotein/apolipoprotein markers of women with cardiovascular disease (CVD) risk. The results obtained at the end of the intervention confirmed our hypothesis, since several changes in these markers were observed as a consequence of the interventions carried out, which may have certain relevance in terms of prevention of CVD.

Previous observational and cohort studies have highlighted the relevance of fat intake at breakfast on cardiovascular disease. A large prospective study observed that replacing 5% of daily energy from saturated fatty acids (SFAs) with an equivalent intake from polyunsaturated fatty acids (PUFAs) reduced by 24% the risk ratio of CVD [29]. A meta-analysis has revised the effect of the 1% substitution of SFA by an equivalent amount of PUFA, observing a decrease in plasma triglycerides, total cholesterol, and low-density lipoprotein (LDL) [30]. Contrary to what was expected, a review has described that incremental postprandial triglycerides were lower with an SFA-rich meal, suggesting that the postprandial duration is of importance when evaluating the effects of fatty acids in plasma lipoproteins [31], which highlights the need for long-term studies like the present one.

Focusing on the effect of the interventions on the different plasma lipid variables, it is important to comment that PUFA and monounsaturated fatty acid (MUFA) breakfasts were associated with a significant decrease of chylomicrons, in spite of the fact that the plasma samples were collected in fasting conditions (12 h), where its concentration was supposed to be low or nonexistent. As a previous report has shown that age is a key regulator of the chylomicron response to the dietary fat [32], age was considered as a covariate in our analysis, confirming our results.

The breakfast containing a PUFA-rich source (margarine) was able to decrease many proatherogenic particles, as very low-density lipoproteins (VLDL) and intermediate-density lipoproteins (IDL), although the effect on LDL was modest. As commented by Aneni et al., lipoprotein subfraction analysis may help to further discriminate patients who require more intensive cardiovascular work-up and treatment [33]. Considering the effect of PUFA-rich breakfasts on the lipoproteins, it can be speculated that this intervention reduced the cardiovascular risk of the participants.

Interestingly, the breakfast containing a source of SFA (butter) was also able to decrease VLDL lipoproteins, but the effect on the other proatherogenic markers (IDL and LDL) was inverse to that observed in the PUFA-rich breakfast. A previous study also showed that replacing butter for margarine improved the blood lipoprotein profile and reduced the predicted risk of coronary heart disease [34], as in the present work.

Although the effect of fat on plasma lipoproteins has been studied in depth, the data regarding the effect of the modification of the type of fat ingested at breakfast on the lipoprotein subclasses is quite scarce. Recent studies have examined whether LDL size distribution or concentration of small LDL (LDL-3, LDL-4, LDL-5, LDL-6, and LDL-7) are strongly related to cardiac event rates (death, myocardial infarction, and revascularization for refractory ischemia) and coronary artery disease (CAD) [6,35,36]. LDL size seems to be an important predictor of cardiovascular events and progression of coronary artery disease, and a predominance of small dense LDL has been accepted as an emerging cardiovascular risk factor by the National Cholesterol Education Program Adult Treatment Panel III [37].

In this study, our data showed that LDL-4 and LDL-5 (small dense LDL) levels decreased with PUFA-rich and MUFA-rich breakfasts, but increased with SFA-rich breakfasts. Although the estimated effect may seem modest, it is important to remember that the relative contribution of these subfractions is small, so a reduction of only 0.4 mg/mL of LDL-4, as observed in the present study in the PUFA-rich intervention, is proportional to a 40% reduction of plasma LDL-4 levels. Previous works have described a similar decrease of small dense LDL particles by PUFA supplementation [38], but it is also important to remember that other nutrients like carbohydrates modulate their plasma levels [39]. As all participants followed the same dietary program, we can speculate that our observations were mainly due to the treatment.

Elevated levels of intermediate density lipoprotein (IDL) are also associated with increased cardiovascular risk [35,40]. The results derived from the present study showed that PUFA-rich and MUFA-rich breakfasts decreased total IDL levels, especially the IDL-c subclass. As IDL-c is the smallest IDL subfraction, its decrease in plasma concentration could also be related to a lower cardiovascular risk [41].

One of the parameters most affected by the different interventions carried out was HDL lipoprotein. PUFA-rich and MUFA-rich breakfasts increased its plasma values, whereas SFA-rich breakfasts exerted the opposite effect. In fact, the only statistically significant effect of MUFA-rich breakfasts on lipoproteins was this increase of HDL, although it is also true that only with this intervention were the total LDL values reduced.

The data derived from apolipoproteins (Apo) were more controversial. The PUFA-rich breakfast was characterized by an increase of ApoA-lipoproteins, while both PUFA-rich and MUFA-rich breakfasts decreased ApoB levels. ApoB is the integral protein of chylomicron and their remnants, VLDL, IDL, LDL, and Lp(a). Results from recent epidemiological studies suggest that ApoB is better than LDL to predict coronary events [42]. In a previous report, Gagliardi et al. examined the effects of butter and margarine and found a significant reduction in ApoB when margarine was used, as in the present work [34]. ApoA-II is the second most abundant HDL protein but its function remains largely unknown. Studies in humans and genetically modified mice have highlighted the stabilizing role of ApoA-II on HDL through inhibition of their remodeling by lipases [43] and through the modulation of LPL activity [44]. An interesting theory of these works is related to the functionality of HDL particles. These studies have revealed the presence of dysfunctional HDL in patients with cardiovascular disease [43,44]; therefore, it can be hypothesized that the determination of ApoA-II is more relevant than that of HDL, especially in patients with cardiovascular disease. Therefore, the effect of the PUFA-rich breakfast on cardiovascular health could be more advantageous than expected, as previously commented [45].

As a whole and considering the data obtained, PUFA and MUFA were associated with a lower coronary risk profile, a situation that has been broadly detailed previously [12,13]. The interest of the present study is, in our opinion, that the lipoprotein profile of a patient can be improved with a simple intervention such as the modification of breakfast fat, and therefore, a sudden change in eating habits, which could hinder adherence to the intervention, is not necessary. Nevertheless, several limitations should be commented on at this point. On the one hand, the three interventions were not isocaloric, nor did they provide the same fat quantity. In this regard, we decided to keep the fat intake provided as monodose sachets and envelopes, the most frequently used by the general Spanish population, which in our opinion would be more suitable for a regular intake. On the other hand, certain clinical parameters could have increased the relevance of the intervention on the patient’s cardiovascular risk, so our observations are limited to the plasma lipoprotein parameters.

## 5. Conclusions

If we consider the modifications of the plasma lipid profile, our data suggest that simply increasing the PUFA or MUFA content at breakfast is enough to decrease Lp(a), VLDL, and IDL and to increase HDL levels, so these types of fats improved the atherogenic lipid profile of the participants. The increase of Apo AI and decrease of Apo B with intake of a PUFA-rich breakfast also suggest that this intervention can prevent the risk of CVD. Although the effect of the PUFA-rich breakfast seems greater, it must be remembered that this breakfast was the one with the least amount of fat, which is undoubtedly also influencing the data obtained. Given the high prevalence of CVD, a simple modification such as changing the fat serving at breakfast may serve to promote health status, especially in those patients with an impaired lipoprotein profile. Nevertheless, further studies with a similar design to the present work are necessary to evaluate whether these interventions effectively reduce cardiovascular risk and/or increase life expectancy of CVD patients.

## Figures and Tables

**Figure 1 nutrients-12-01695-f001:**
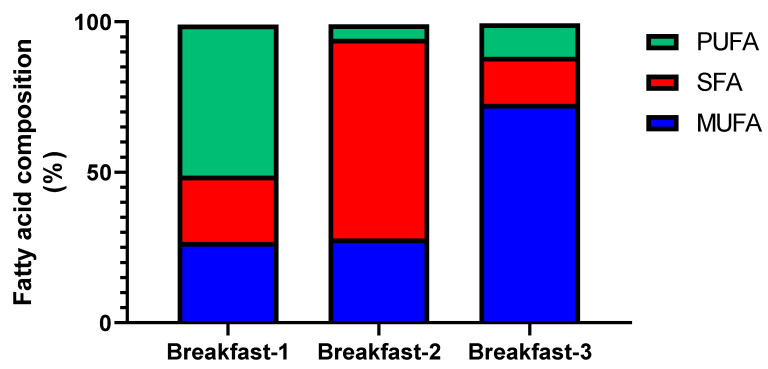
Fatty acid composition (in fat grams per serving) of the different interventions (based on the manufacturer’s information). Breakfast-1 was mainly composed of PUFA, breakfast-2 of SFA, and breakfast-3 of MUFA.

**Figure 2 nutrients-12-01695-f002:**
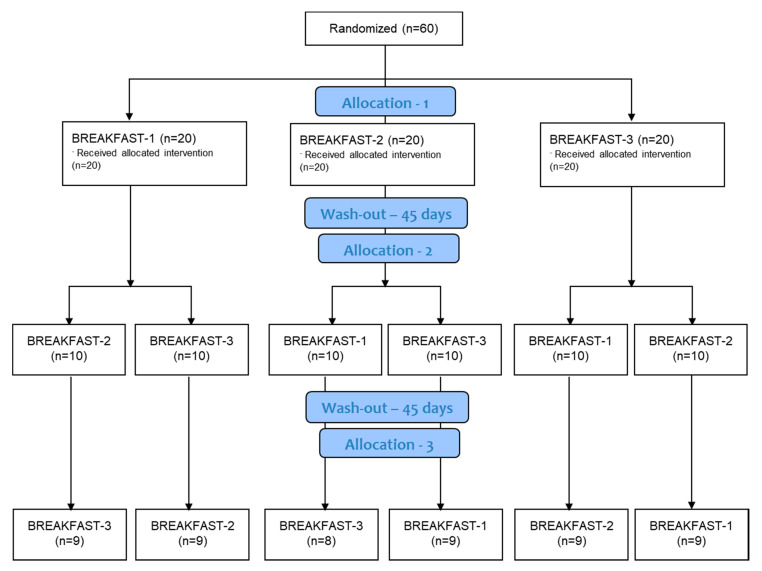
Participants’ flow diagram of the present study.

**Figure 3 nutrients-12-01695-f003:**
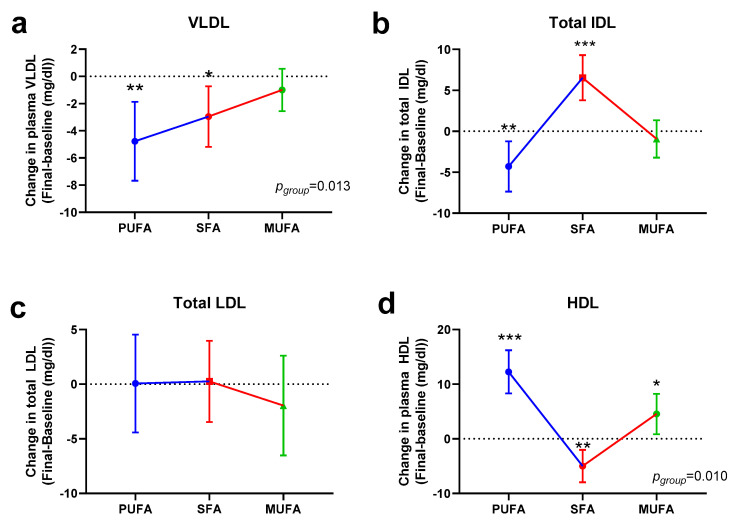
Changes in very low-density lipoproteins (VLDL, **a**), intermediate-density lipoproteins (IDL, **b**), low-density lipoproteins (LDL, **c**), and high-density lipoproteins (HDL, **d**) after 30 days of intervention with the different breakfasts. Data represent estimated treatment differences (final-baseline) values and the 95% confidence interval (CI). Data derive from those women who completed the study (*n* = 53). * represents differences within the same group (final-baseline). * *p* < 0.050, ** *p* < 0.010, *** *p* < 0.001. Within-group differences are indicated as *p*_group_. Further information is available in Appendix A.

**Figure 4 nutrients-12-01695-f004:**
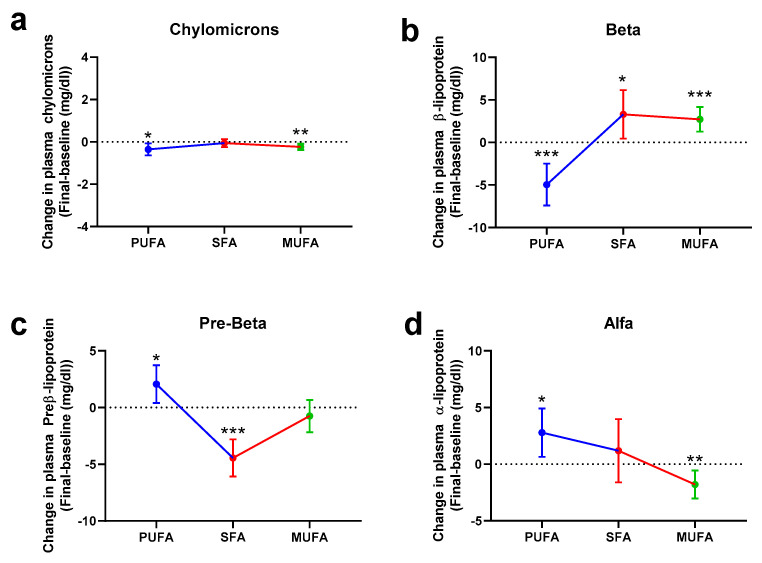
Changes in chylomicrons (**a**), beta (**b**), pre-beta (**c**), and alfa (**d**) lipoprotein fractions attending to their mobility after 30 days of intervention with the different breakfasts. Data represent estimated treatment differences (final-baseline) values and the 95% confidence interval (CI). Data derive from those women who completed the study (*n* = 53). * represents differences within the same group (final-baseline). * *p* < 0.050, ** *p* < 0.010, *** *p* < 0.001. No statistically significant difference was observed among groups. Further information is available at Appendix A.

**Figure 5 nutrients-12-01695-f005:**
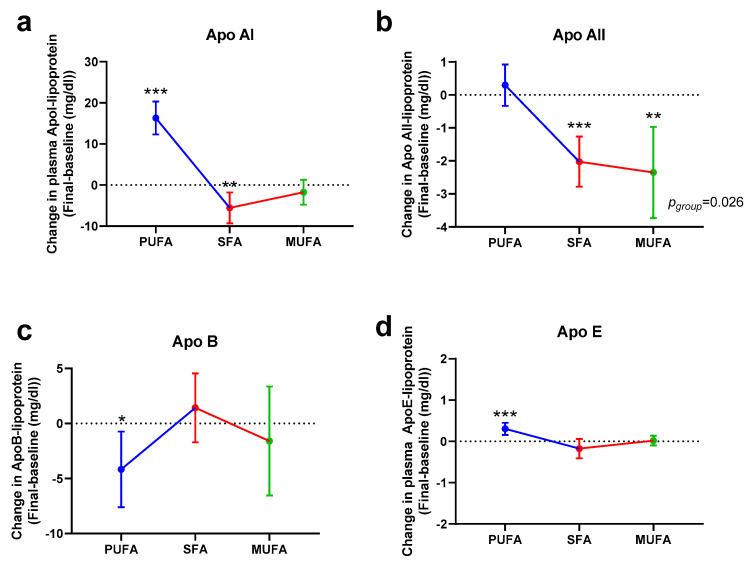
Changes in plasma apolipoproteins (Apo) AI (**a**), Apo AII (**b**), Apo B (**c**), and Apo E (**d**) after 30 days of intervention with the different breakfasts. Data represent estimated treatment differences (final-baseline) values and the 95% confidence interval (CI). Data derive from those women who completed the study (*n* = 53). * represents differences within the same group (final-baseline). * *p* < 0.050, ** *p* < 0.010, *** *p* < 0.001. Within-group differences are indicated as *p*_group_. Further information is available at Appendix A.

**Table 1 nutrients-12-01695-t001:** Baseline characteristics of the studied population.

	Baseline Characteristics (*n* = 53)
Age (years)	63.5 ± 18.4 [31.2–85.4]
Weight (kg)	64.6 ± 8.9 [52.8–78.5]
Body Mass Index	27.79 ± 3.97 [21.75–34.09]
SBP (mmHg)	13.0 ± 2.2 [10.2–17.1]
DBP (mmHg)	7.3 ± 1.0 [6.0–9.0]
Personal History of CVD (*n*,%)	9, 17%
Family History of CVD (*n*,%)	17, 32%
Diabetes History (*n*,%)	7, 13%
VLDL (mg/dL)	24.7 ± 9.2 [22.0–27.4]
IDL (mg/dL)	42.2 ± 8.4 [39.8–44.6]
LDL (mg/dL)	65.0 ± 12.7 [61.3–68.7]
HDL (mg/dL)	62.9 ± 15.4 [58.4–67.4]
Apo AI lipoprotein (mg/dL)	174.4 ± 21.1 [168.2–180.6]
Apo AII lipoprotein (mg/dL)	35.2 ± 5.3 [33.6–36.7]
Apo B lipoprotein (mg/dL)	87.4 ± 16.1 [82.6–92.1]
Apo E lipoprotein (mg/dL)	4.4 ± 0.9 [4.1–4.6]

Data are mean ± standard deviation; 95% confidence interval is described in brackets. CVD: cardiovascular disease. DBP: diastolic blood pressure. HDL: high-density lipoproteins. IDL: intermediate-density lipoproteins. LDL: low-density lipoproteins. SBP: systolic blood pressure. VLDL: very low-density lipoproteins.

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
