# Peer review of "Effect of the Fat Eaten at Breakfast on Lipid Metabolism: A Crossover Trial in Women with Cardiovascular Risk"

_nutrients, 2020, doi:10.3390/nu12061695_

Round 1

Reviewer 1 Report

Review regards the paper entitled

Effect of the fat eaten at breakfast on lipid 2 metabolism: a crossover-trial in women with  cardiovascular risk”.

The manuscript was constructed correctly and the results presented are of scientific value.

In my opinion, there are several issues that require additional clarification. Below are also some suggestions that the authors should consider.

  1. Please check all abbreviations. I mean if all given abbreviations have got their full names? (example page 1, line 39 ApoB). First should be full name, and next the abbreviation.
  2. Please extend the description of fats (margarine, butter and olive). Was the determination of the content of fatty acids carried out (GC determination)?? Or given information are based on the manufacturer's information? Whether this margarine had partially hydrogenated fat or interesterified fat? How do the authors know that margarine did not contain trans isomers ?? In Table S1, the sum of the fatty acids for each fat should be 100%. If it is otherwise, a change in the table signature is required, e.g. “selected fatty acids”. Did margarine contain other functional additives, i.e. stanols?
  3. In my opinion, Figure 1 needs improvement. Since the amount of fat was always the same, why the bars are different??
  4. The study ran out of blank test. The study should also include a group of people who did not consume any portion of fat for breakfast at all.
  5. Page 3, line 115 Figure 1 – should be Figure 2.
  6. Has the proposed breakfast diet always been the same (excluding fat) each day?
  7. Page 4 line 134, What did authors mean?? – “Although fat content was slightly different….”
  8. In my opinion the resolution of Figure2 is poor.
  9. Please supply the legend into the Figures 3,4,5…a,b,c,d ..need explanation.., besides, the records such as: Bet, Pre Beta et al. should be written in full or their abbreviation should be provided as legend (Figures 4 and 5)

Author Response

Reviewer #1.- Comments and Suggestions for Authors

The manuscript was constructed correctly and the results presented are of scientific value. In my opinion, there are several issues that require additional clarification. Below are also some suggestions that the authors should consider.

We would like to thank the reviewer’s comments. We have tried to follow his/her indications as detailed below:

#1.- Please check all abbreviations. I mean if all given abbreviations have got their full names? (example page 1, line 39 ApoB). First should be full name, and next the abbreviation.

We regret the lack of information. We have included abbreviation information throughout the revised version of the paper.

#2.- Please extend the description of fats (margarine, butter and olive). Was the determination of the content of fatty acids carried out (GC determination)?? Or given information are based on the manufacturer's information?

As the reviewer comments, the description of fat content was based on manufacturer’s information. This issue has been commented in the revised version of the paper and in Table S1.  

#3.- Whether this margarine had partially hydrogenated fat or interesterified fat?  How do the authors know that margarine did not contain trans isomers ??

At first, we assumed that margarine was free of hydrogenated, intersterified or trans isomers (apart from those occurring naturally). A previous report of the Spanish Federation of Nutrition and Dietetic (FESNAD) has confirmed that the margarine employed in the present work was absent of these types of fat. [1]. But most important, attending to the European regulations (available at: https://eur-lex.europa.eu/legal-content/EN/TXT/HTML/?uri=CELEX:32019R0649&from=ES), food intended for the final consumer and food intended for supply to retail, shall not contain trans fat (apart from the naturally occurring fat of animal origin).

  1. Salas, J.; Romero, M.; Villarino, A. Consenso Sobre Las Grasas Y Aceites En La Alimentación. Fed. Española Soc. Nutr. Aliment. y Dietética 2007, 80.

#4.- In Table S1, the sum of the fatty acids for each fat should be 100%. If it is otherwise, a change in the table signature is required, e.g. “selected fatty acids”. Did margarine contain other functional additives, i.e. stanols?

We completely agree with the reviewer and have modified Table S1 accordingly. On the other hand, this margarine did not contain other functional additives, again attending to the manufacturer's information.

#5.- In my opinion, Figure 1 needs improvement. Since the amount of fat was always the same, why the bars are different??

Effectively, we regret this misunderstanding. We have modified the original figure to show the data more appropriately.

#6.- The study ran out of blank test. The study should also include a group of people who did not consume any portion of fat for breakfast at all.

The comment of the reviewer is fairly interesting and, in fact, it was an important debate before starting this work. Finally, we decided that the wash-out period, considering each participant as their own control, since the washout out was enough to return analytical parameters to their baseline values.  Therefore, the breakfast established in the washing periods allowed us to consider each participant as part of that control group, since in the washout-breakfast they did not eat any type of fat with the bread toast, as in the other interventions. This procedure has been also employed in several previous studies [2,3]

  1. Keogh, G.F.; Cooper, G.J.S.; Mulvey, T.B.; McArdle, B.H.; Coles, G.D.; Monro, J.A.; Poppitt, S.D. Randomized controlled crossover study of the effect of a highly β-glucan-enriched barley on cardiovascular disease risk factors in mildly hypercholesterolemic men. Am. J. Clin. Nutr. 2003, 78, 711–718, doi:10.1093/ajcn/78.4.711.
  2. Samkani, A.; Skytte, M.J.; Anholm, C.; Astrup, A.; Deacon, C.F.; Holst, J.J.; Madsbad, S.; Boston, R.; Krarup, T.; Haugaard, S.B. The acute effects of dietary carbohydrate reduction on postprandial responses of non-esterified fatty acids and triglycerides: A randomized trial. Lipids Health Dis. 2018, 17, 1–9, doi:10.1186/s12944-018-0953-8.

#7.- Page 3, line 115 Figure 1 – should be Figure 2.

It has been corrected.

#8.- Has the proposed breakfast diet always been the same (excluding fat) each day?

Effectively. In fact, we consider that this is one of the main points of the present work. Since all women were in the same institution, we were able to control the intake of the same breakfast throughout the study period.

#9.- Page 4 line 134, What did authors mean?? – “Although fat content was slightly different….”

We made reference that although the serving portions were similar (10g/serving), the water content of margarine and butter is higher than of olive oil, so fat content is slightly different. This aspect has been commented in the revised version of the paper.

#10.- In my opinion the resolution of Figure2 is poor.

We completely agree with the reviewer. We have supplied a new version of the figure to increase clearness.

#11.- Please supply the legend into the Figures 3,4,5…a,b,c,d ..need explanation.., besides, the records such as: Bet, Pre Beta et al. should be written in full or their abbreviation should be provided as legend (Figures 4 and 5)

We regret this lack of information. It has been modified in the revised version of the paper.

In summary, we would like to thank the reviewer’s comments. We considered that the changes made following the indications have significantly improved the quality of the paper.

Juan José Hernández Morante on behalf of all my co-authors.

Reviewer 2 Report

  • Authors are suggested to mention the brands of fats which they used.
  • Authors are suggested to explain in detail. If they ran the samples of all lipoprotein fractions on PAGE, they might have calculated Rf values; if those were calculated then data of fractions should be representing difference in travel distance (Rf). Authors are suggested to check this and mention in detail how they did. Did they run all 53 members pre and post study samples? If so, please provide representative images 
  • Did the authors compared the page subfractions with lipoprotein levels from blood/plasma using “cholestech” or any other device which can measure directly; since in methodology it was mentioned that the samples were pre-stained/labeled which might impact the movement on PAGE. Methodology is little confusing. Please write the methodology in detail how did they label or stain the lipoproteins first itself prior running gel and with what agent.
  • Authors also suggested to explain in briefly about the lipidogram in agarose gel and immune-nephelometry.

Author Response

Reviewer #2.- Comments and Suggestions for Authors

#1.- Authors are suggested to mention the brands of fats which they used.

The fat sources were selected based on two parameters: that they offer single-dose serving portions with the amount of product that was planned in the study design and that they were the most consumed brands in our environment. This information was obtained from https://acude.org/omic-murcia/ and https://www.datacentric.es/

Then, the selected brands were:

Butter: Rioba® (purchased in Makro®.  Further information at: https://distribucion-hosteleria.makro.es/shop/pv/BTY-X297534/0032/0021/Mantequilla-pura-sin-sal-RIOBA-150-monodosis-10g)

Margarine: Flora® (purchased in Alcampo®. Further information at: https://www.flora.es/productos/margarinas/flora-original)

Virgin Olive oil: Capricho Andaluz ® (purchased online at: https://caprichoandaluz.com/?v=57f5037281be. Further information at: https://caprichoandaluz.com/productos/monodosis-aove/monodosis-aove-tarrinas/caja-aceite-v-e-552-unid-x-10-ml/?v=57f5037281be).

However, since none of these brands had any involvement in the design or in any other part of the study, we did not consider it appropriate to include this information in the article. Nevertheless, if the reviewer still considers it necessary, we would include such information in a further revision of the paper.

#2.- Authors are suggested to explain in detail. If they ran the samples of all lipoprotein fractions on PAGE, they might have calculated Rf values; if those were calculated then data of fractions should be representing difference in travel distance (Rf). Authors are suggested to check this and mention in detail how they did. Did they run all 53 members pre and post study samples? If so, please provide representative images

We apologize for the lack of information at this regard. Following the reviewer’s indications, we have detailed this methodology in the revised paper. In addition, the following references have been included to further clarify this section:

  1. Tsui, A.K.Y.; Thomas, D.; Hunt, A.; Estey, M.; Christensen, C. Lou; Higgins, T.; Sandhu, I.; Rodriguez-Capote, K. Analytical sensitivity and diagnostic performance of serum protein electrophoresis on the HYDRAGEL 30 PROTEIN(E) β1-β2 Sebia Hydrasys system. Clin. Biochem. 2018, 51, 80–84, doi:10.1016/j.clinbiochem.2017.09.007.
  2. Weinstock, N.; Bartholome, M.; Seidel, D. Determination of apolipoprotein A-I by kinetic nephelometry. Biochim. Biophys. Acta (BBA)/Lipids Lipid Metab. 1981, 663, 279–288, doi:10.1016/0005-2760(81)90214-9.
  3. Muñiz, N. Measurement of plasma lipoproteins by electrophoresis on polyacrylamide gel. Clin. Chem. 1977, 23, 1826–33.

Otherwise, as the reviewer comments, the 53 patients followed this procedure. Unfortunately, as these procedures were carried out in a Hospital facility, all samples were destructed following Spanish regulations. Nonetheless, we have included examples from other study as a representative sample.

Figure 1. Lipoprotein subfraction distributions for five individuals - from a homogeneous LDL pattern on the left to a progressively more heterogeneous pattern on the right.

Figure 2. Typical Lipoprint profile of a subject with normal values of plasma lipoproteins.

Figure 3. Typical Lipoprint profile of a subject with abnormal values of plasma lipoproteins.

#3.- Did the authors compared the page subfractions with lipoprotein levels from blood/plasma using “cholestech” or any other device which can measure directly; since in methodology it was mentioned that the samples were pre-stained/labeled which might impact the movement on PAGE. Methodology is little confusing. Please write the methodology in detail how did they label or stain the lipoproteins first itself prior running gel and with what agent.

The comment of the reviewer is fairly interesting. Effectively, sample handling was conducted with an automated system. In line with the previous comment, we regret the lack of information in the original paper. We have tried to detail this methodology in the revised version of the paper.

#4.- Authors also suggested to explain in briefly about the lipidogram in agarose gel and immune-nephelometry.

We have modified the Methods section of the paper to increase information about these procedures. We sincerely regret the lack of information in the previous version of the paper.

In summary, we would like to thank the reviewer’s comments. We considered that the changes made following the indications have significantly improved the quality of the paper.

Juan José Hernández Morante on behalf of all my co-authors.
